# SURDS: Benchmarking Spatial Understanding and Reasoning in Driving Scenarios with Vision Language Models

**Xianda Guo**[1,*], **Ruijun Zhang**[2,3,*], **Yiqun Duan**[4,*], **Yuhang He**[5], **Dujun Nie**[2,3],
**Wenke Huang**[1], **Chenming Zhang**[6,3], **Shuai Liu**[7], **Hao Zhao**[8], **Long Chen**[2,3,6,†]

[1] School of Computer Science, Wuhan University
[2] Institute of Automation, Chinese Academy of Sciences     [3] Waytous
[4] University of Technology Sydney     [5] Microsoft Research
[6] IAIR, Xi'an Jiaotong University     [7] ByteDance     [8] AIR, Tsinghua University
xianda_guo@163.com; {zhangruijun2023, long.chen}@ia.ac.cn; duanyiquncc@gmail.com

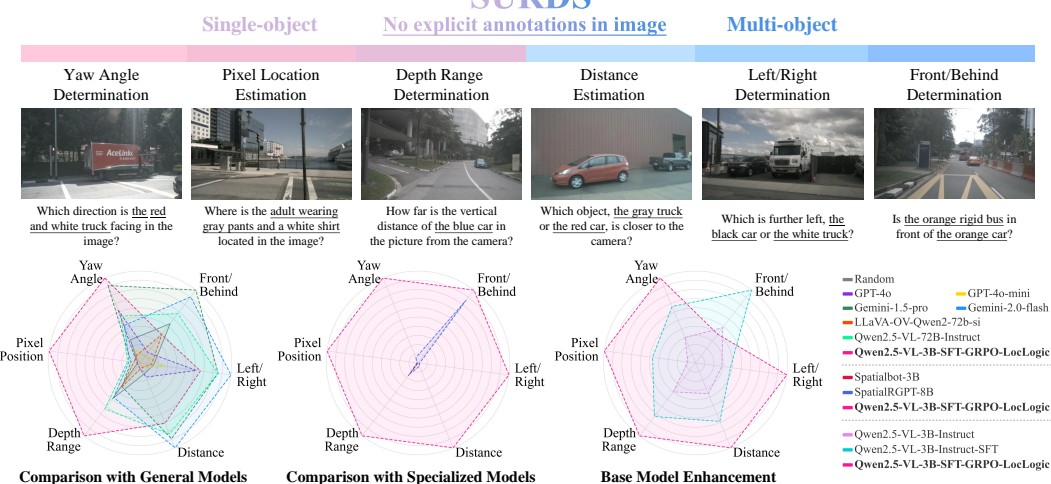

Figure 1: **Overview of the SURDS benchmark and the proposed method's performance.** The upper part illustrates the SURDS benchmark, which comprises six challenging tasks within driving scenarios, divided into single-object and multi-object categories. The lower part presents three radar charts evaluating the proposed method: the bottom left compares its performance against large-scale open-source and proprietary models; the bottom center highlights comparisons with existing spatial understanding methods; and the bottom right shows successive enhancements from the base model.

## Abstract

Accurate spatial reasoning in outdoor environments—covering geometry, object pose, and inter-object relationships—is fundamental to downstream tasks such as mapping, motion forecasting, and high-level planning in autonomous driving. We introduce SURDS, a large-scale benchmark designed to systematically evaluate the spatial reasoning capabilities of vision language models (VLMs). Built on the NUSCENES dataset, SURDS comprises **41,080** vision–question–answer training instances and **9,250** evaluation samples, spanning six spatial categories:

---

[*]These authors contributed equally to this work.

[†]Corresponding author

39th Conference on Neural Information Processing Systems (NeurIPS 2025) Track on Datasets and Benchmarks.

orientation, depth estimation, pixel-level localization, pairwise distance, lateral ordering, and front–behind relations. We benchmark leading general-purpose VLMs, including GPT, Gemini, and Qwen, revealing persistent limitations in fine-grained spatial understanding. To address these deficiencies, we go beyond static evaluation and explore whether alignment techniques can improve spatial reasoning performance. Specifically, we propose a reinforcement learning–based alignment scheme leveraging spatially grounded reward signals—capturing both perception-level accuracy (*location*) and reasoning consistency (*logic*). We further incorporate final-answer correctness and output-format rewards to guide fine-grained policy adaptation. Our GRPO-aligned variant achieves overall score of 40.80 in SURDS benchmark. Notably, it outperforms proprietary systems such as GPT-4o (13.30) and Gemini-2.0-flash (35.71). To the best of our knowledge, this is the first study to demonstrate that reinforcement learning–based alignment can significantly and consistently enhance the spatial reasoning capabilities of VLMs in real-world driving contexts. We release the SURDS benchmark, evaluation toolkit, and GRPO alignment code through: `https://github.com/XiandaGuo/Drive-MLLM`.

# 1   Introduction

Understanding complex spatial structures—such as object orientation, relative position, and geometric layout—within either discrete images or sequential videos [8] serves as a fundamental challenge in embodied perception and multi-modal scene understanding. Such spatial reasoning abilities are essential for a wide range of downstream tasks, including motion prediction [29, 86, 55, 32, 60], planning [65, 7, 49] and map construction [79, 19]. While we have witnessed huge progress in per-object centered recognition tasks with the assistance of various large-scale dataset [21, 30, 58], including detection [38, 39], tracking [42, 24], optical flow estimation [22, 57], depth estimation [37, 34–36] and semantic segmentation [16, 40], the inter-object spatial relation reasoning from RGB images in autonomous driving has been largely ignored even despite its vital importance in achieving fully holistic 3D scene understanding.

In the vision community, spatial relation reasoning within a single image has received increasing attention [68] by leveraging datasets such as Visual Genome [44]. However, existing works primarily focus on simple 2D positional relations (e.g., left/right, top/bottom), which fail to capture the richness and address the ambiguity of 3D spatial dependencies critical on real-world environments. Meanwhile, the recent emergence of large language models (LLMs) [61, 75] and their multimodal variants (VLMs) [1, 52, 15, 78] has opened promising avenues for high-level vision-language reasoning. Yet, the extent to which these models can perform on spatial understanding remains unclear.

Despite recent efforts on spatial question answering, such as BLINK [27], SpatialBot [11], Spatial-RGPT [11], most prior studies focus on controlled indoor environments or rely on auxiliary modules such as depth estimators or object detectors. These designs limit generalization to dynamic and visually complex scenes. In contrast, spatial reasoning is required in outdoor scenarios, especially driving scenarios. At the same time, a growing number of works have begun to directly employ LLMs for planning or decision-making in autonomous driving [73, 80, 59, 23], yet such approaches often overlook a key prerequisite: without first establishing the spatial reasoning capability of these models, deploying them for real-world driving actions is inherently unreliable. This highlights the urgent need for a dedicated benchmark to systematically evaluate VLMs' spatial understanding in driving contexts.

In this work, we propose to systematically evaluate and improve the spatial reasoning capabilities of VLMs via a new large-scale benchmark: SURDS. Built on the nuScenes [10] dataset, SURDS comprises multi-view driving scenes captured from six surrounding cameras. From this, we respectively curate 41,080 training and 9,250 validation vision–question–answer (VQA) samples designed to probe fine-grained spatial understanding across six dimensions: orientation, depth, pixel-level position, pairwise distance, lateral ordering, and front–behind relations. Each query is paired with linguistically diverse, contextually grounded questions and evaluated using task-specific metrics. We compare the proposed SURDS benchmark with existing spatial understanding benchmarks in Table 1. SURDS is the first spatial understanding benchmark in driving scenarios.

Table 1: **Comparison between our work and other spatial understanding benchmarks.** Scale denotes the total number of QA pairs in the benchmark. Annotation Types indicate how spatial relations are labeled. Reasoning indicates whether the framework focuses on reasoning. Method denotes whether it proposes a specific approach to enhance spatial understanding. w/o Depth means the framework does not use depth information during training or evaluation. w/o Visual Mark indicates no visual annotations are added to the image.

| Paper | Scale | Annotation Types | Data Source | Reasoning | Method | w/o Depth | w/o Visual Mark |
|---|---|---|---|---|---|---|---|
| BLINK (ECCV2024) [27] | 3,807 | Image QA pairs | Web | ✗ | ✗ | ✓ | w/ Marked point |
| SpatialRGPT (NeurIPS2024) [18] | 1,406 | Image QA pairs | Web | ✓ | ✓ | ✗ | w/ Mask |
| SpatialBot (ICRA2025) [11] | 174 | Image QA pairs | Web | ✓ | ✓ | ✗ | w/ Marked point & Bbox |
| VSI bench (CVPR2025) [81] | ~5,000 | Video QA pairs | Indoor | ✓ | ✗ | ✓ | ✓ |
| **SURDS (ours)** | 9,250 | Image QA pairs | Driving | ✓ | ✓ | ✓ | ✓ |

Using SURDS, we systematically evaluate the spatial reasoning capabilities of several frontier VLMs, including GPT-4o, GPT-4o-mini, Gemini, and Gemini 2.5 Pro. Our analysis reveals four converging lines of evidence that existing models still lack robust spatial grounding. Our evaluation reveals that current VLMs, including large-scale models, surprisingly struggle with spatial reasoning, showing poor absolute localization and brittle multi-object relational understanding abilities. Meanwhile, if the performance the spatial performance grows accordingly as the model scales up still remains unclear.To explore potential improvements, we first scale up synthetic spatial data via agent-centric scene construction (see Figure 3), which yields a baseline with an overall accuracy of 26.94 on our benchmark. Building on top of this, we propose a specially designed perception with reasoning process reward as shown in Figure 3 and together with the original final answer reward and format reward, we employ Group Relative Policy Optimization towards reasoning-level signals. Our GRPO-aligned model further boosts the performance to 40.80, not only reaching SoTA compared with the models at the same scale but also surpassing the most advanced large models such as GPT-4o (13.30), Gemini-2.0-flash (35.71), and Qwen2.5-VL-72B (33.47). Our experiments reveal the current spatial understanding ability for most existing SoTA works. Our benchmark exposes critical limitations in the spatial reasoning capabilities of large models within driving scenarios, and we further demonstrate that reinforcement learning–based alignment can substantially enhance these abilities. To facilitate future research, we release the comprehensive dataset, evaluation toolkit, and alignment pipeline that offer hands-on resources for advancing grounded spatial understanding in VLMs.

- We propose SURDS, the first large-scale benchmark for evaluating fine-grained spatial understanding of VLMs in realistic driving scenarios, respectively, contains 41,080 training pairs and 9,250 test pairs.

- Our evaluations on SURDS reveal fundamental spatial reasoning limitations in existing models and demonstrate that model scale alone does not ensure spatial competence.

- Comprehensive experiments with different training strategies ranging from supervised fine-tuning, reinforcement learning to post-train alignment, provide valuable insight for follow-up researchers.

## 2 Related Work

**Large Vision Language Models (LVLM)** Benefiting from the huge success in large language models (LLMs) [9, 61, 62] in recent years, a new research venue has been focusing on extending natural language-based large models (especially the GPT family LLM) to multimodal large language models (VLM) [1, 47, 48, 66, 74, 25]. Among all of them, encompassing vision into language has made dramatic progress and various vision language models (VLM) have been developed [5, 6, 47, 48, 51, 52] for various crossmodal tasks such as visual question answer (VQA) [3, 31] and crossmodal reasoning [84, 41, 28, 71], owing to the availability of various large image-text datasets [50, 43, 69, 12]. Typical VLM models include BLIP family [47, 48], LLaVA family [51, 52] and Qwen-VL family [4, 5, 76]. They either innovate in network architecture [17, 64, 47, 48] or adopt novel training strategy [5, 87]. For example, regarding the network architecture innovation, QWen-VL [5] and MiniGPT-4 [87] employ ViT [2] like network as visual encoder, LLaVA [64] instead employs CLIP ViT-L/14 [67] for visual encoding and InternVL [17] uses InternViT-6B for

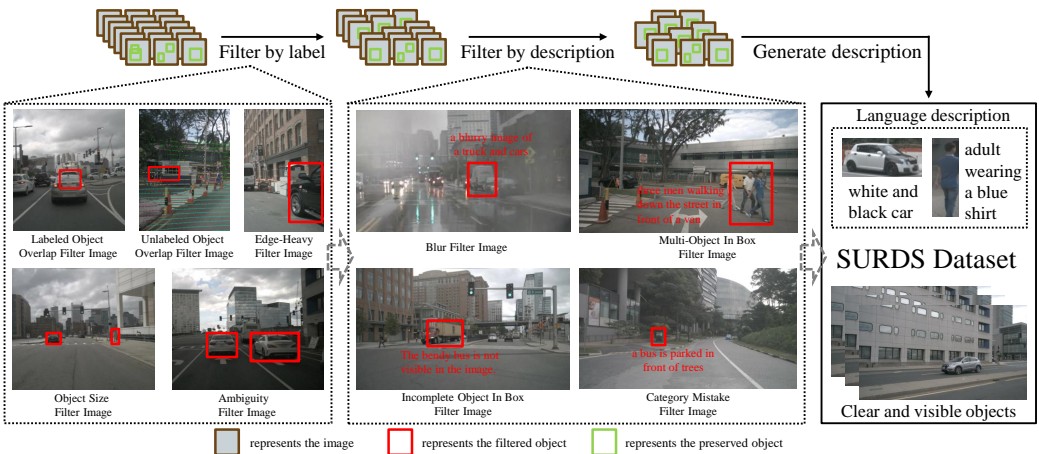

Figure 2: **Overview of the pipeline for constructing the SURDS dataset.** The visual elements shown in the image (e.g., bounding boxes and textual descriptions) are illustrative only and do not appear in the actual dataset. The system filters objects based on labels (left), applies additional filtering using text descriptions generated by a vision language model (center), and produces object descriptions (right), ensuring high-quality annotations for each image.

visual encoding. Regarding the training strategy, Qwen-VL [5] employs a three-stage strategy: first pre-train on massive image-text pairs, then multi-task pre-train over seven major tasks, and finally fine-tune with instruction on over 350,000 dialogues. MiniGPT-4 [87] adopts a two-stage training strategy by first pre-training on composite dataset including Conceptual Captions [13], LAION [70] and SBU [63] and then fine-tuning on high-quality image description dataset.

**Visual-Language Benchmarks**

In the pre-LLM era, most public vision-language datasets were single-task oriented, limiting their ability to holistically evaluate multimodal reasoning. Representative examples include image captioning [50], visual question answering [3, 31], and OCR [54]. With the emergence of LLMs, more comprehensive and multi-task datasets have been curated to better assess general-purpose multimodal reasoning. Among them, MME [26] focuses on *Yes/No* questions, visual perception, and language reasoning; MMBench [53] expands coverage across diverse domains with a circular evaluation design; Seed-Bench [46, 45] introduces multi-image and video inputs; and MM-Vet [82] aggregates multiple sub-tasks, including OCR, recognition, and math reasoning. Beyond recognition-centric benchmarks, recent efforts target broader cognitive abilities. MMMU [83] emphasizes domain knowledge reasoning, HallusionBench [33] investigates hallucinations and visual illusions, MathVista [56] focuses on math-based visual understanding, BLINK [27] probes holistic perception, and Mega-Bench [14] scales evaluation to 500+ real-world tasks.

## 3   SURDS Benchmark

Recent advancements have seen VLMs being directly employed for autonomous driving and embodied intelligence, which heavily depend on sophisticated spatial perception and reasoning. However, these works lack a detailed investigation into the spatial reasoning abilities of VLMs to demonstrate how reliable current models are on spatial information.

**Data Source**   We construct our benchmark on the nuScenes [10] dataset, which is a large-scale public dataset specifically designed for autonomous driving research. It collects rich sensor data, including images from six cameras covering a full 360° field of view, along with LiDAR, radar, and GPS/IMU data. The dataset is captured in the urban environments of Boston and Singapore, featuring a diverse range of traffic conditions, weather scenarios, and times of day. This diversity ensures that the models are tested on a wide array of real-world driving situations, enhancing the robustness of the evaluation.

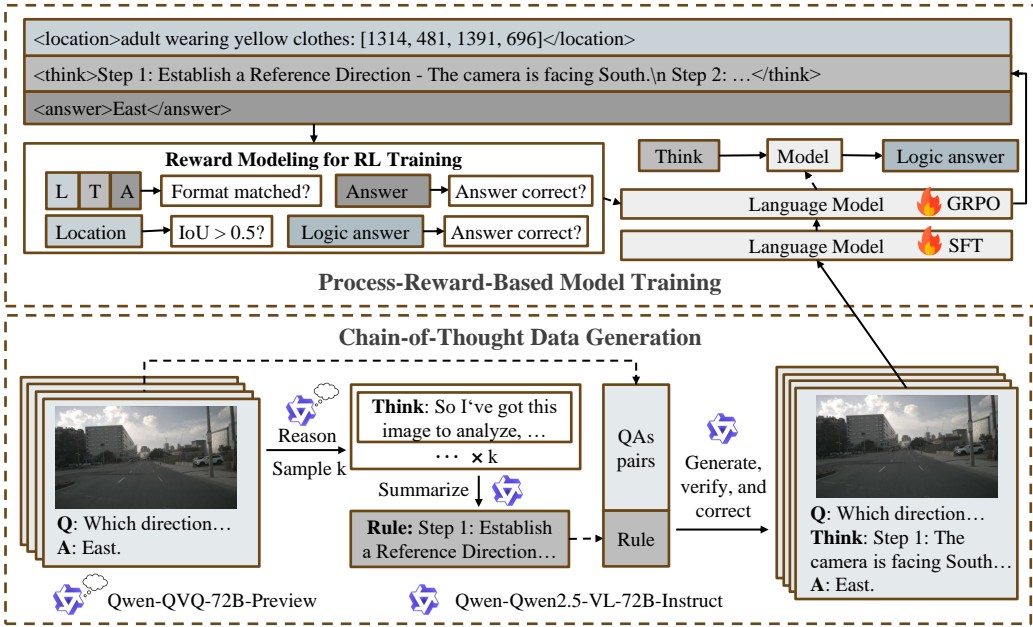

Figure 3: **Overview of data generation and model training.** The lower part illustrates how COT-augmented QA data is generated: sampled QA pairs are processed by a vision reasoning model to infer solutions, followed by a vision-language model that extracts general reasoning steps and rules. These rules are used to generate COT-augmented QA pairs, which are then validated and corrected. The upper part shows the training pipeline: after SFT, the model is further optimized using process-reward-based training guided by four custom rewards.

**Data Filtering**    To ensure that each bounding box corresponds to a clearly visible and unambiguous object, we adopt a multi-stage filtering pipeline comprising both label-based and description-based strategies, as shown in Figure 2. These filters collectively remove occluded, edge-aligned, ambiguous, and undersized objects:

- **Occlusion Removal.** We discard objects that are heavily occluded by other annotated instances when the ratio $r = \frac{|A \cap B|}{\min(|A|,|B|)}$, where $A$ and $B$ denote the areas of the two bounding boxes, exceeds 0.8. To further account for occlusion caused by unlabeled obstacles (e.g., fences), we project the *lidarseg* point clouds from nuScenes onto the image plane and remove any object whose bounding box contains too few LiDAR points of the correct semantic class, suggesting that the object is not meaningfully visible.

- **Edge and Size Filters.** We exclude objects whose center points lie outside image boundaries, as well as those with pixel areas below a minimal size threshold. To prevent identity ambiguity, we also remove images containing multiple objects of the same class (e.g., several pedestrians or vehicles).

- **Description-based Filtering.** Even after geometric filtering, some objects remain semantically ambiguous due to blur or annotation noise. We use *instruction-blip-13B* to generate object-level descriptions and discard any instance producing unclear or non-specific text. For retained samples, we standardize references to reduce linguistic bias—vehicles are described solely by color (e.g., "white and black car"), while pedestrians are identified by clothing (e.g., "adult wearing a blue shirt").

Starting from 28,130 training and 6,019 validation multi-view scenes (6 cameras per scene), we retain 27,152 and 5,919 images with clearly visible objects, respectively. These form the foundation for our QA benchmark, yielding 41,080 training and 9,250 validation vision–question–answer (VQA) instances.

**Benchmark Construction**    To systematically evaluate VLMs' spatial reasoning capabilities in realistic driving contexts, we construct a large-scale QA benchmark consisting of 41,080 training

and 9,250 validation vision–question–answer (VQA) instances. These are derived from filtered images captured by six cameras in the nuScenes dataset. We design six spatial tasks across two categories. The single-object subset focuses on basic spatial comprehension—yaw angle classification, pixel-level localization, and depth range estimation—each probing a distinct axis of object-centric reasoning. The multi-object subset introduces relational reasoning, including pairwise distance comparison, left–right ordering, and front–back understanding, requiring the model to analyze spatial relationships across multiple objects. The prompt templates formulated for various VQA tasks are provided in Appendix B.

## 4 Post-Train Alignment for Boosting Spatial Understanding

### 4.1 Data Generation by Test-Time CoT Scaling

RL training primarily enhances a VLM's performance based on existing knowledge [20], while SFT functions serves as a process of knowledge injection. We argue that models with smaller parameter sizes lack inherent reasoning capabilities and thus require SFT to introduce reasoning knowledge. Since the constructed dataset does not contain chain-of-thought (CoT) annotations, it is necessary to generate reasoning traces for each QA pair.

To provide more insights for the community, we employ open-source models to generate CoTs. As shown at the bottom side of Figure 3, we first use the visual reasoning model QVQ to reflect on and reason through a sampled set of k QA pairs. These thought processes are then summarized and distilled into reasoning rules consisting of generalizable solution steps using Qwen2.5-VL-72B. By feeding these rules alongside the original QA pairs into Qwen2.5-VL-72B, we generate new QA pairs annotated with CoTs. These outputs are subsequently validated and corrected by the model itself, resulting in an automated pipeline for constructing high-quality QA datasets with COT annotations. The prompts used to generate CoT reasoning for the data are provided in Appendix D. This method is motivated by two empirical observations: 1) Directly generating CoTs in batch using QVQ is computationally expensive and often results in verbose, unstructured, or format-inconsistent outputs; 2) Relying solely on Qwen2.5-VL-72B for CoT generation leads to degraded output quality and increased hallucinations.

### 4.2 Reinforcement Training with Reward Modeling

After obtaining the CoT-augmented data, we use it to train the model to enhance its spatial reasoning capabilities. The training pipeline is illustrated at the top side of Figure 3. We begin with **SFT as the cold start** of the full model, including the training of the visual encoder, multimodal projector, and language model, using the generated long CoT data.

Then, we apply reinforcement learning using GRPO [72] to enhance the model's capacity for spatial reasoning. In each training instance, the model is prompted to sequentially generate three components: the bounding box of the queried object, a step-by-step reasoning trace, and the final answer. Given that spatial reasoning is inherently object-centric, we assign a localization reward of 1 if the predicted bounding box achieves an IoU greater than 0.5 with the ground-truth region. To encourage output fidelity, a format reward of 1 is given if the model adheres to the prescribed output structure. Additionally, an accuracy reward of 1 is granted when the final answer is correct.

To promote logical coherence in reasoning, we introduce a logic reward inspired by Embodied-R [85], which assesses whether the reasoning trace leads to a correct answer. While the original approach uses a frozen reference model to evaluate consistency by feeding in both the reasoning trace and the question, we identify two limitations: (1) including the question can introduce answer leakage, as the trace often implicitly encodes the answer; and (2) relying on a static external model entails a trade-off between inference cost and reliability. To address both issues, we feed only the reasoning trace into the model under training, which serves as its own verifier. This design is efficient, incurs no additional computational overhead, and dynamically adapts to the model's evolving capabilities. A scalar logic reward of 1 is assigned if the inferred answer from the reasoning trace matches the originally generated final answer; otherwise, the reward is 0. This combination of localization, format, accuracy, and logic rewards ensures that the model not only produces structurally valid outputs, but also grounds its reasoning process in spatial consistency.

Table 2: **Comparison of our proposed method with other open-source and proprietary VLMs, as well as specialized spatial understanding models.** Yaw, Pixel, Depth, Dis, L/R, and F/B correspond to the six spatial reasoning tasks illustrated in Figure 1. The Score column represents the average performance across these six metrics. **Bold**: Best. Underline: Second Best.

| Model | Single-object | | | Multi-object | | | Score |
|---|---|---|---|---|---|---|---|
| | Yaw | Pixel | Depth | Dis | L/R | F/B | |
| Random | 5.73 | 1.12 | 34.27 | 8.76 | 11.57 | 11.89 | 12.22 |
| GPT-4o | 13.08 | 1.62 | 2.49 | 11.57 | 47.89 | 3.14 | 13.30 |
| GPT-4o-mini | 3.24 | 0.28 | 0.22 | 4.22 | 21.51 | 2.05 | 5.25 |
| Gemini-1.5-pro | 19.14 | 4.41 | 22.70 | 61.95 | 66.38 | **22.05** | 32.77 |
| Gemini-2.0-flash | 9.30 | 5.41 | 32.97 | **69.30** | **77.30** | 20.00 | 35.71 |
| LLaVA-OV-Qwen2-72b-si | 1.95 | 3.03 | 23.57 | 3.78 | 9.73 | 8.65 | 8.45 |
| Qwen2.5-VL-72B-Instruct | 11.57 | 6.13 | 44.00 | 58.05 | 66.16 | 14.92 | 33.47 |
| Qwen2.5-VL-7B-Instruct | 7.57 | 3.46 | 25.95 | 11.46 | 17.95 | 9.30 | 12.61 |
| Qwen2.5-VL-3B-Instruct | 6.27 | 3.81 | 27.68 | 17.84 | 14.81 | 10.49 | 13.48 |
| SpatialBot [11] | 0.00 | 0.00 | 12.00 | 0.00 | 0.00 | 0.00 | 2.00 |
| SpatialRGPT [18] | 1.30 | 0.55 | 10.59 | 1.95 | 0.86 | 7.35 | 3.77 |
| Qwen2.5-VL-3B-SFT-GRPO-LocLogic | **20.97** | **44.81** | **69.84** | 49.30 | 51.35 | 8.54 | **40.80** |

# 5 Experiments

## 5.1 Experimental Setup

**Implementation Details** We assess a variety of models, including state-of-the-art open-source and proprietary models, as well as models specifically designed for spatial understanding. A random baseline is also included for comparison, and the evaluated models are summarized in Table 2. All models are prompted with standardized instructions and are required to generate outputs strictly adhering to a predefined format. We employ the `sglang` framework[*] to accelerate inference and reduce evaluation time. All training and evaluation are conducted on eight NVIDIA A800 GPUs. For supervised fine-tuning, models are trained for 2 epochs with a learning rate of $1 \times 10^{-6}$ and a warm-up ratio of 10%. GRPO training is performed for 1 epoch using a maximum prompt length of 4096 tokens, an output length of up to 512 tokens, and generating 4 samples per prompt. The structured response format is provided in Appendix C.

**Evaluation Metrics**

To quantitatively evaluate model performance on the SURDS benchmark, we define a set of evaluation metrics. For the *Pixel Localization Estimation* task, we adopt a centerness-based [77] metric. For other tasks, a prediction receives a score of 1 if it matches the ground-truth answer, and 0 otherwise. Given $N$ QA pairs, the metric score for each task is computed as the average over all $N$ pairs. The final overall score is the average of all individual task scores.

To avoid misleading results from scalar regression metrics, we adopt a range-based categorical evaluation rather than predicting precise numeric values. This design better reflects human-like spatial reasoning: for example, a person can usually tell whether a car is facing north, south, east, or west, but cannot quote its exact yaw angle from a single image. In addition, this categorical scheme prevents models from exploiting trivial solutions such as always predicting an average scalar value. Overall, the dual-query, range-based evaluation provides a more robust and honest assessment of fine-grained spatial reasoning ability.

## 5.2 Main Results

The evaluation results of different models are presented in Table 2. Among proprietary models, Gemini performed the best, achieving top results across several multi-object metrics. Among open-source VLMs, Qwen also showed strong performance, ranking second in many metrics. The specialized spatial understanding model performed poorly across all tasks, while the SpatialBot model could not be evaluated on several metrics due to its lack of instruction-following capability. Our proposed method achieved first place in multiple single-object metrics, with a significant margin over the second-best model—for instance, a nearly 60% improvement on the depth metric. It also

---

[*]`https://github.com/sgl-project/sglang`

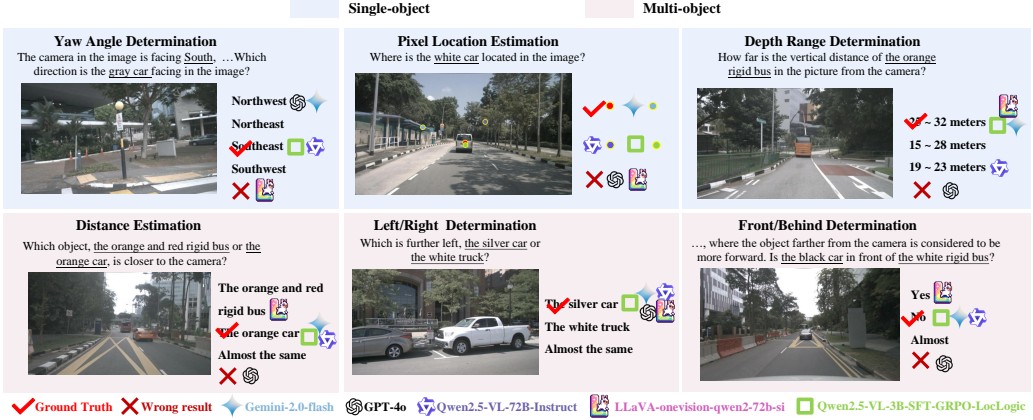

Figure 4: **Illustrative examples of the benchmark QA pairs on both single-object and multi-object.**

achieves the highest overall score, outperforming the second-best approach by 14.25%. We visualize example responses from different models across various tasks, as shown in Figure 4. It can be seen that our proposed model performs well across multiple tasks. Furthermore, we visualize the complete outputs of our proposed model across various tasks, demonstrating its strong reasoning capabilities, as illustrated in Appendix A.

## 5.3 Discussion on Benchmarking Performance SOTA models

Here, we discuss the quantitative results of our benchmark with respect to four key aspects: single-object evaluation, multi-object evaluation, the relationship between model size and performance, and the impact of fine-tuning. Our benchmark offers three converging lines of evidence that current open-source VLMs still lack robust spatial grounding. **1) single-object probes** (yaw, pixel coordinates, depth bins) reveal that most models—including several 70 B-parameter variants—perform at or below random chance on orientation and depth, and seldom exceed 10% accuracy in sub-pixel localization, underscoring a persistent inability to encode absolute pose or metric information. **2) multi-object** tests show a modest uptick in accuracy for simpler comparative questions (left–right ordering, pairwise distance), yet performance collapses when the task demands non-canonical reasoning such as identifying the object "in front" under a forward-facing reference frame, indicating that relative spatial heuristics remain brittle. **3) scaling analysis** demonstrates that parameter count is not a reliable predictor of spatial competence: larger models sometimes trail lighter counterparts, implying that mere capacity expansion without explicit geometric priors does little to close the reasoning gap.

## 5.4 Ablation Study

**Ablation of reward.** We conducted ablation studies on our proposed method. The first set of experiments focused on the composition of the reward. Since the basic GRPO framework inherently includes the format reward and accuracy reward, we do not ablate these two. Instead, we examine the effects of adding or removing the location reward and logic reward. As shown in Table 3, incorporating either the location or logic reward individually led to only marginal improvements in overall performance. However, when both rewards were applied together, the model experienced a significant performance boost. These results suggest that without the supervision provided by the location reward, the model's object localization ability degrades—this ability forms the foundation of spatial reasoning. Building on this, the inclusion of logic supervision further enhances the model's consistency and spatial reasoning capability.

The second set of experiments focuses on the value settings of the rewards. We compared the conventional reward setting of 0, 1 with an alternative setting of -1, 1. The results show that using 0, 1 yields better performance. We attribute this to the sparsity of rewards during training—if the model receives a penalty every time it fails to reason correctly, it would accumulate mostly negative

Table 3: **Ablation study on model performance under different reward settings after SFT.** The four types of rewards correspond to the representations illustrated in Figure 3. †means using -1 and 1 as binary rewards instead of 0 and 1.

| Base model | Training reward | | | | Single-object | | | Multi-object | | | Score |
|---|---|---|---|---|---|---|---|---|---|---|---|
| | Format | Loc | Acc | Logic | Yaw | Pixel | Depth | Dis | L/R | F/B | |
| Qwen2.5-VL-3B-SFT | ✗ | ✗ | ✗ | ✗ | 13.95 | 21.11 | 51.35 | 33.95 | 19.68 | 21.62 | 26.94 |
| Qwen2.5-VL-3B-SFT | ✓ | ✗ | ✓ | ✗ | 19.24 | 15.02 | 62.59 | 39.14 | 32.65 | 9.30 | 29.66 |
| Qwen2.5-VL-3B-SFT | ✓ | ✓ | ✓ | ✗ | 17.84 | 22.72 | 64.65 | 41.41 | 30.92 | 11.68 | 31.53 |
| Qwen2.5-VL-3B-SFT | ✓ | ✗ | ✓ | ✓ | 20.54 | 14.81 | 62.49 | 36.54 | 32.65 | 11.35 | 29.73 |
| Qwen2.5-VL-3B-SFT | ✓ | ✓ | ✓ | ✓ | 20.97 | 44.81 | 69.84 | 49.30 | 51.35 | 8.54 | **40.80** |
| Qwen2.5-VL-3B-SFT† | ✓ | ✓ | ✓ | ✓ | 18.16 | 22.61 | 62.05 | 39.14 | 32.22 | 15.14 | 31.55 |
| Qwen2.5-VL-3B-SFT | ✓ | ✓ | ✓ | ✓ | 20.97 | 44.81 | 69.84 | 49.30 | 51.35 | 8.54 | **40.80** |

Table 4: **Ablation study on different training setups of our method, analyzing the impact of SFT, GRPO, and the addition of location and logic rewards (LocLogic).** SFT denotes supervised fine-tuning, GRPO stands for Group Relative Policy Optimization.

| Model | Single-object | | | Multi-object | | | Score |
|---|---|---|---|---|---|---|---|
| | Yaw | Pixel | Depth | Dis | L/R | F/B | |
| Qwen2.5-VL-3B | 6.27 | 3.81 | 27.68 | 17.84 | 14.81 | 10.49 | 13.48 |
| Qwen2.5-VL-3B-GRPO | 14.59 | 3.75 | 29.19 | 35.68 | 39.89 | 22.70 | 24.30 |
| Qwen2.5-VL-3B-GRPO-LocLogic | 8.11 | 57.82 | 27.24 | 22.05 | 19.35 | 12.43 | 24.50 |
| Qwen2.5-VL-3B-SFT | 13.95 | 21.11 | 51.35 | 33.95 | 19.68 | 21.62 | 26.94 |
| Qwen2.5-VL-3B-SFT-GRPO | 19.24 | 15.02 | 62.59 | 39.14 | 32.65 | 9.30 | 29.66 |
| Qwen2.5-VL-3B-SFT-GRPO-LocLogic | 20.97 | 44.81 | 69.84 | 49.30 | 51.35 | 8.54 | **40.80** |

rewards. This causes gradients to remain negative or close to zero over extended periods, thereby hindering learning progress.

**Ablation of training** We also conducted ablation studies on the training of the base model. Specifically, we compared the standard GRPO training setup (using only format reward and accuracy reward) with our proposed reward design that incorporates format reward, location reward, accuracy reward, and logic reward. Similarly, we applied the same training configurations to the SFT-pretrained model. The results are presented in Table 4. Across both the non-SFT and SFT models, GRPO training improves overall performance. However, for the non-SFT model, adding the location and logic rewards did not lead to further improvements. In contrast, for the SFT model, incorporating location and logic rewards resulted in a significant performance boost. We attribute this difference to the weaker localization ability of the non-SFT model. Due to this limitation, the location reward remains sparse and provides limited training benefit, leading to negligible performance gains.

# 6   Conclusion and Limitation Discussion

We introduce SURDS, a large-scale benchmark comprising 41,080 VQA training instances and 9,250 evaluation samples that span six spatial reasoning categories—orientation, depth estimation, pixel-level localization, pairwise distance, lateral ordering, and front–behind relations. Benchmarking state-of-the-art general-purpose VLMs on SURDS exposes persistent shortcomings in fine-grained spatial understanding. To mitigate these deficiencies, we propose a reinforcement-learning framework that integrates spatially grounded reward signals with a reasoning-consistency objective. Extensive comparative and ablation experiments demonstrate that our approach yields substantial performance gains over existing VLMs while empirically validating the efficacy of both the training method and the reward design.

Nonetheless, the method remains untested on larger-scale model variants, and the benefits of linear reward scaling and multi-stage GRPO schedules have yet to be clarified. In addition, our QA construction primarily targets perception-level spatial reasoning. While these tasks naturally underpin higher-level capabilities such as trajectory prediction and planning, SURDS does not yet include such downstream QA pairs, which we leave for future work. Overall, we believe SURDS and its accompanying alignment strategy provide a solid foundation for advancing spatial reasoning in VLMs and open promising directions toward safer and more capable autonomous driving systems.

**Acknowledgements.** This work was supported by the Joint Funds of the National Natural Science Foundation of China under U24B20162 and the National Natural Science Foundation of China under Grant (62373356, 623B2080).

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

# Supplementary Material on SURDS

## A Example Illustrating the Reasoning Process

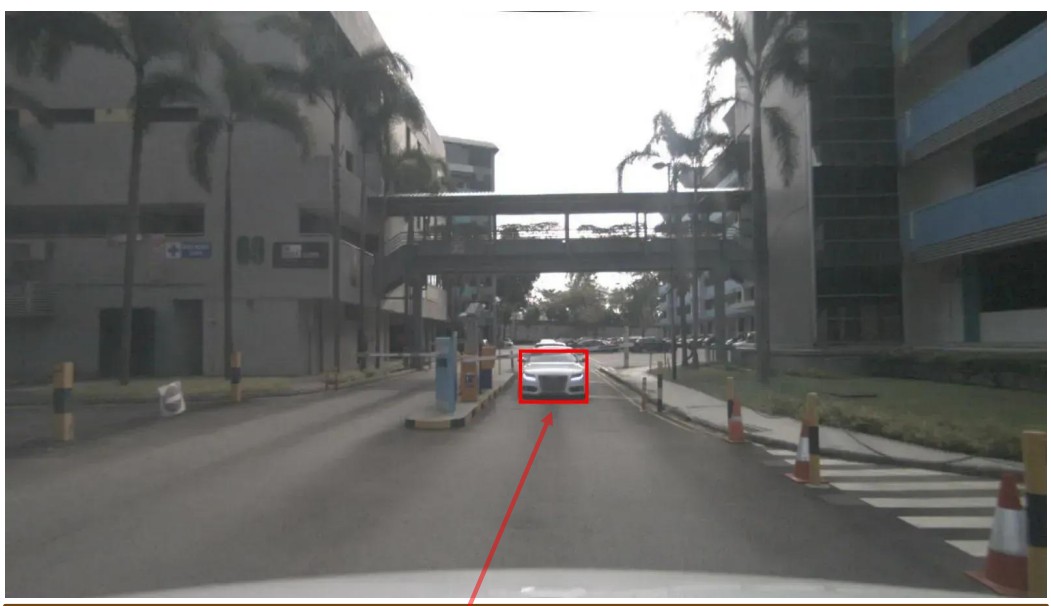

**Question**:
Task Description:
The primary goal of this task is to identify the direction that the specified object is facing in the given image. **The camera in the image is facing North**, and you need to analyze the object's orientation based on this reference.
Question: **Which direction is the white car facing in the image?**
Options: \n- North\n- South\n- East\n- West

First, detect and list the target objects mentioned in the question from the image. Output their bounding boxes under the <location> tag. Next, reason carefully and step-by-step under the <think> tag to ensure logical accuracy and robustness, including any relevant error checks. Finally, provide a concise and definitive response in the <answer> tag. Use the following format: <location>[[Detected target object]: [xmin, ymin, xmax, ymax], ...]</location> <think>[Step-by-step reasoning with attention to detail and potential error checks]</think><answer>[Final answer]</answer>

**Answer**:
<location>white car: **[789, 519, 895, 601]**</location>
<think>
Step 1: Establish a Reference Direction
**The camera is facing North, so this is our baseline direction.**
Step 2: Determine Object Position Relative to Camera
The white car is positioned in front of the camera on the road, **moving towards the camera.**
Step 3: Analyze Object Orientation Based on Reference and Position
**Since the car is moving towards the camera and the camera is facing North, the car must be facing South.**
Step 4: Consider Additional Contextual Clues
There are no visible contextual clues such as road markings or traffic signs that would suggest a different orientation for the car.
Step 5: Verify Consistency with Given Options
The derived orientation of the car facing South is consistent with the given options.
Step 6: Formulate a Clear Final Answer
After verifying the analysis, the final answer is clear.
</think>
<answer>**South**</answer> ☑

Figure 5: Example of Yaw Angle Determination task.

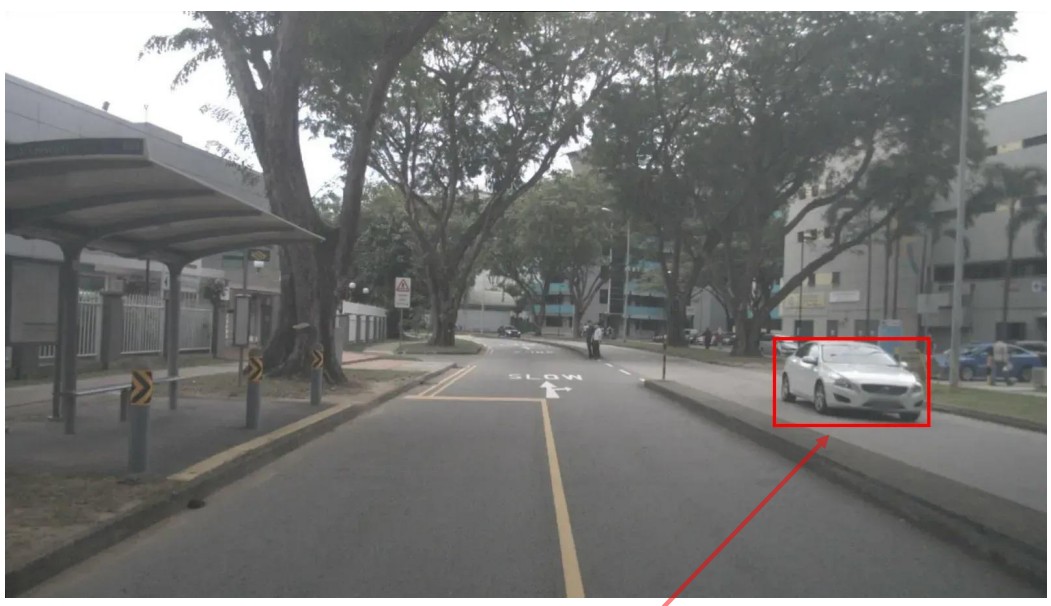

**Question**:
Task Description:
The primary goal of this task is to accurately identify and provide the coordinates of a specified object within a given image. Your task is to analyze the image, locate the object, and return its position in the form of coordinates [x, y].
Question: **Where is the white car located in the image?**
First, detect and list the target objects mentioned in the question from the image. Output their bounding boxes under the <location> tag. Next, reason carefully and step-by-step under the <think> tag to ensure logical accuracy and robustness, including any relevant error checks. Finally, provide a concise and definitive response in the <answer> tag. Use the following format: <location>[[Detected target object]: [xmin, ymin, xmax, ymax], ...]</location>
<think>[Step-by-step reasoning with attention to detail and potential error checks]</think><answer>[Final answer]</answer>

**Answer:**
<location>white car: **[1179, 500, 1420, 637]**</location>
<think>
Step 1: The task is to identify the coordinates of the white car in the image.
Step 2: **The image shows a white car on the right side of the road, close to the curb.**
Step 3: The white car is the object of interest, and its position should be identified.
Step 4: **Estimating the coordinates, the white car appears to be around the middle-right of the image. The x-coordinate is approximately 1320, and the y-coordinate is around 551.**
Step 5: Verifying the solution, the coordinates seem plausible as they place the car near the middle-right of the image.
Step 6: No alternative scenarios are considered as the car is clearly visible and identifiable.
Step 7: The final answer is [1320, 551].
Step 8: The assumptions made are based on the visual cues and the car's position relative to the image boundaries.
</think>
<answer>**[1320, 551]**</answer> ☑

Figure 6: Example of Pixel Location Estimation task.

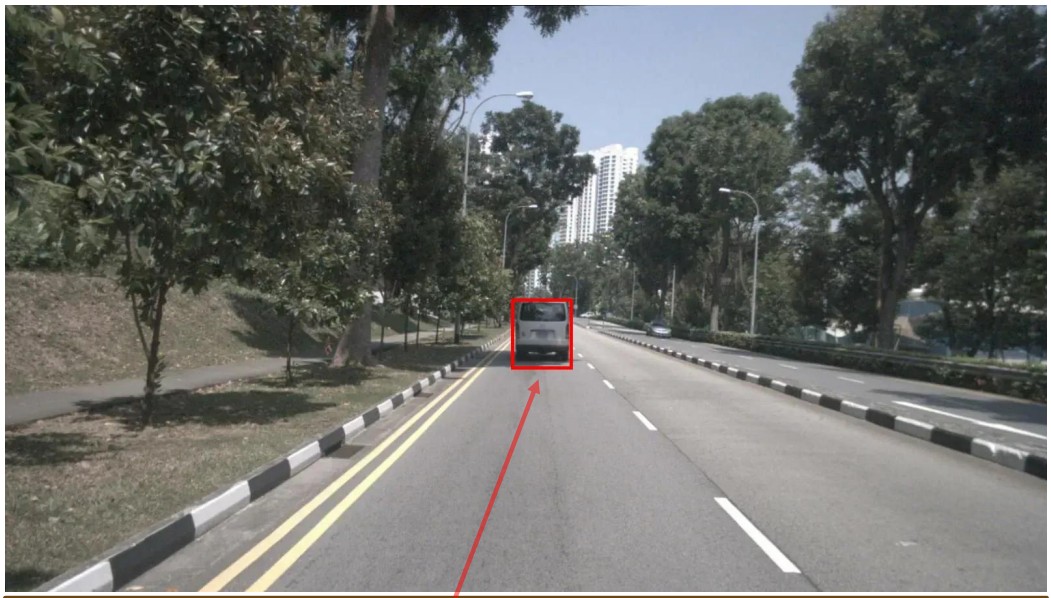

**Question**:
Task Description:
The primary goal of this task is to estimate the vertical distance of the specified object in the image from the camera, which is positioned at the origin. You need to analyze the image and choose the correct range of distance from the camera based on the visual cues provided.
Question: **How far is the vertical distance of the white car in the picture from the camera?**
Options:- Between 15 meters and 18 meters - Between 19 meters and 23 meters - Between 24 meters and 31 meters
First, detect and list the target objects mentioned in the question from the image. Output their bounding boxes under the <location> tag. Next, reason carefully and step-by-step under the <think> tag to ensure logical accuracy and robustness, including any relevant error checks. Finally, provide a concise and definitive response in the <answer> tag. Use the following format: <location>[[Detected target object]: [xmin, ymin, xmax, ymax], ...]</location> <think>[Step-by-step reasoning with attention to detail and potential error checks]</think><answer>[Final answer]</answer>

**Answer:**
<location>white car: **[777, 450, 871, 558]**</location>
<think>
Step 1: Understand the Context and Question
The question asks for the vertical distance of the white car from the camera, which is positioned at the origin. The options provided are ranges of distances in meters.
Step 2: Identify Visual Cues and Reference Points
The image shows a road with trees on both sides and a white car in the distance. The car appears to be a standard-sized vehicle, and the road markings can be used as reference points.
Step 3: Apply Principles of Perspective and Scaling
The white car appears smaller due to perspective, indicating it is farther away. The road markings can help estimate the distance. The car is positioned near the midpoint of the road, which suggests it is not extremely far away.
Step 4: Consider Environmental and Contextual Clues
The environment is a wide road with trees and a clear sky. The car is driving away from the camera, and the road appears to be a typical urban or suburban road.
Step 5: Make Educated Guesses Based on Reasonable Assumptions
**Assuming the car is a standard-sized vehicle and using the road markings as a reference, the car appears to be between 24 meters and 31 meters away from the camera.**
Step 6: Verify and Refine Estimates
Cross-checking the estimate with the given options, the range of 24 meters to 31 meters seems reasonable based on the visual cues and the typical dimensions of a standard-sized car.
Step 7: Select the Most Appropriate Option
Based on the analysis, the most appropriate option is \"Between 24 meters and 31 meters.\"
</think>
<answer>**Between 24 meters and 31 meters**</answer> ✅

Figure 7: Example of Depth Range Determination task.

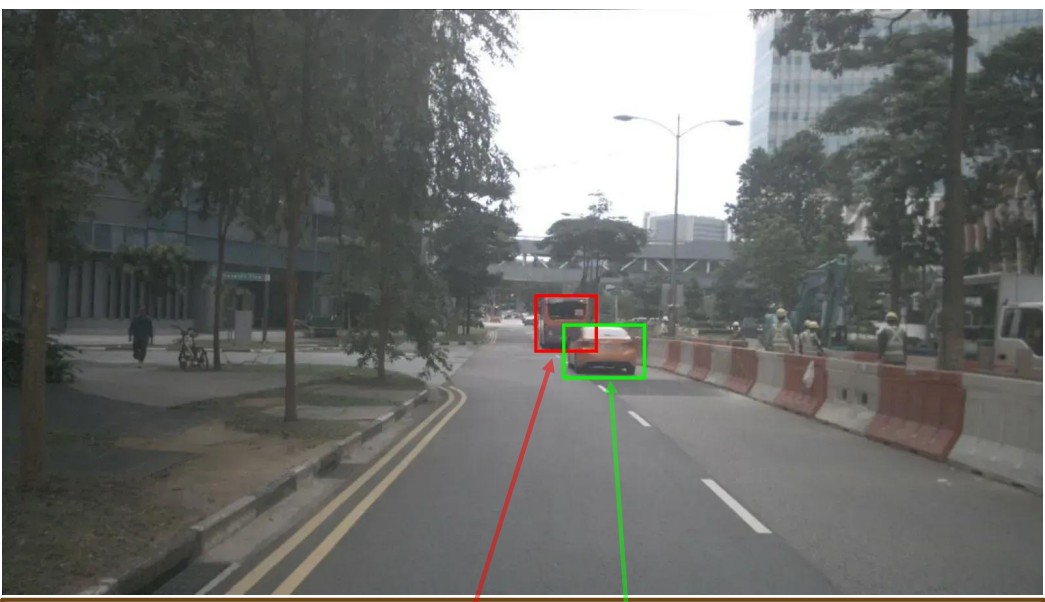

**Question**:
Task Description:
The primary goal of this task is to determine which of the two objects is closer to the camera that captured the image below. You need to assess the relative distance between the two objects based on the camera's perspective.
Question: **Which object, the orange rigid bus or the orange car, is closer to the camera?**
Options: - The orange rigid bus - The orange car - Almost the same
First, detect and list the target objects mentioned in the question from the image. Output their bounding boxes under the <location> tag. Next, reason carefully and step-by-step under the <think> tag to ensure logical accuracy and robustness, including any relevant error checks. Finally, provide a concise and definitive response in the <answer> tag. Use the following format: <location>[[Detected target object]: [xmin, ymin, xmax, ymax], ...]</location> <think>[Step-by-step reasoning with attention to detail and potential error checks]</think><answer>[Final answer]</answer>

**Answer:**
<location>orange rigid bus: **[818, 438, 915, 527]**, orange car: **[861, 483, 989, 568]**</location>
<think>
1. **Identify Scene and Objects**: The scene is a road with an orange rigid bus and an orange car visible. **The bus is farther down the road, and the car is closer to the camera.**
2. **Consider Size and Perspective**: **The bus appears smaller compared to the car. Since objects closer to the camera appear larger, this suggests the car is closer to the camera.**
3. **Evaluate Clarity and Detail**: The car appears more detailed and sharper compared to the bus, which is further away and less detailed.
4. **Analyze Positioning and Overlap**: **The car is positioned closer to the camera and partially overlaps the bus, indicating it is nearer to the camera's viewpoint.**
5. **Examine Depth Cues**: **Linear perspective shows that the road narrows into the distance, and the bus is closer to this narrowing point, suggesting it is farther away.**
6. **Consider Lighting and Shadows**: The lighting and shadows on both objects are consistent with the car being closer to the camera.
7. **Visualize Camera Perspective**: Visualizing from the camera's perspective confirms that the car is closer, as it is larger, more detailed, and positioned closer in the frame.
8. **Cross-Verify Observations**: All observations consistently support the car being closer to the camera.
9. **Select the Best Answer**: Based on the analysis, the most appropriate answer is \"The orange car.\"
</think>
<answer>**The orange car**</answer> 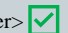

Figure 8: Example of Distance Estimation task.

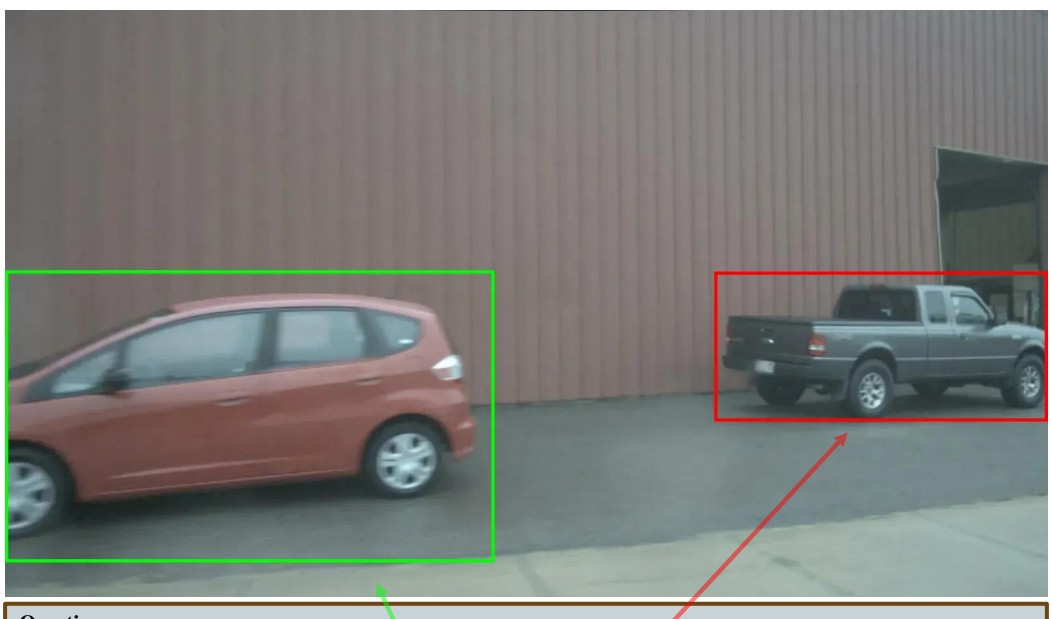

**Question**:
Task Description:
The primary goal of this task is to determine the relative left-right positioning of the two objects from the camera's perspective.
Question: **Which is further right, the gray truck or the red car?**
Options: - The gray truck - The red car - Almost the same
First, detect and list the target objects mentioned in the question from the image. Output their bounding boxes under the <location> tag. Next, reason carefully and step-by-step under the <think> tag to ensure logical accuracy and robustness, including any relevant error checks. Finally, provide a concise and definitive response in the <answer> tag. Use the following format: <location>[[Detected target object]: [xmin, ymin, xmax, ymax], ...]</location> <think>[Step-by-step reasoning with attention to detail and potential error checks]</think><answer>[Final answer]</answer>

**Answer:**
<location>gray truck: **[1088, 401, 1599, 630]**, red car: **[0, 399, 750, 846]**</location>
<think>
Step 1: The image shows a gray truck and a red car parked near a building.
Step 2: The camera perspective is from a slightly elevated angle, looking towards the vehicles.
Step 3: **The gray truck is positioned to the right of the red car relative to the camera's perspective.**
Step 4: **Comparing the positions, the gray truck is clearly further to the right compared to the red car.**
Step 5: **There are no significant depth or perspective effects that would distort the apparent left-right positioning.**
Step 6: The analysis aligns with the visual evidence, confirming the gray truck is further right.
Step 7: Based on the analysis, the gray truck is the correct answer.
</think>
<answer>**The gray truck**</answer> ✅

Figure 9: Example of Left/Right Determination task.

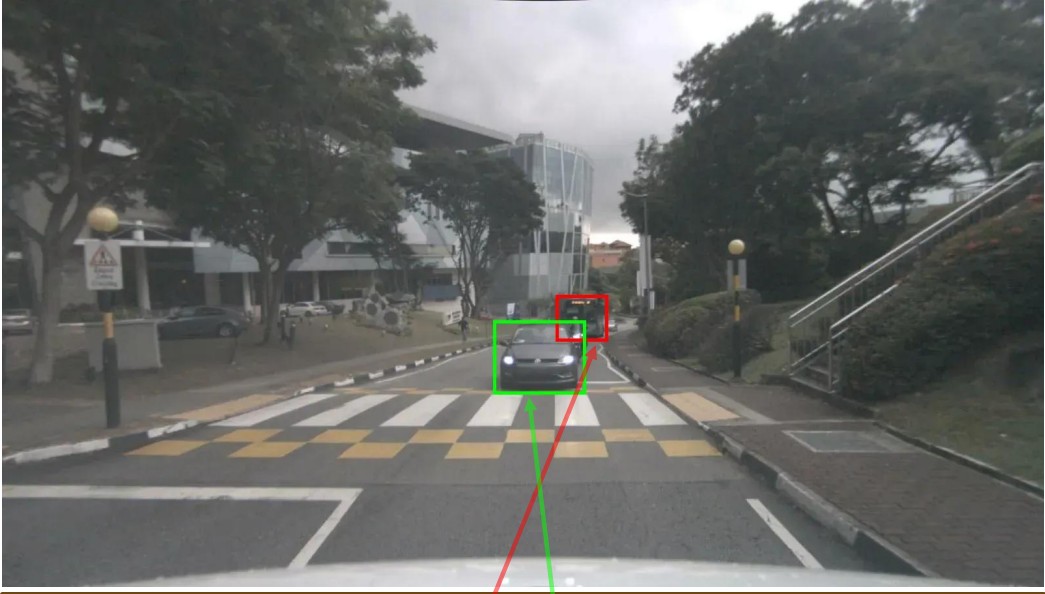

**Question**:
Task Description:
The primary goal of this task is to determine the relative front-back positioning of the two objects from the camera's perspective, **where the object farther from the camera is considered to be more forward**.
Question: **Is the blue rigid bus behind the gray car?**
Options: - Yes – No - Almost the same in terms of front-back position
First, detect and list the target objects mentioned in the question from the image. Output their bounding boxes under the <location> tag. Next, reason carefully and step-by-step under the <think> tag to ensure logical accuracy and robustness, including any relevant error checks. Finally, provide a concise and definitive response in the <answer> tag. Use the following format: <location>[[Detected target object]: [xmin, ymin, xmax, ymax, ...]</location> <think>[Step-by-step reasoning with attention to detail and potential error checks]</think><answer>[Final answer]</answer>

**Answer:**
<location>blue rigid bus: **[850, 452, 930, 523]**, gray car: **[754, 491, 896, 604]**</location>
<think>
Step 1: Identify the Key Objects and Their Descriptions
- The key objects are the blue rigid bus and the gray car.
- **The blue rigid bus is visible in the distance, further down the road.**
- **The gray car is closer to the camera, positioned on the road in front of the bus.**
Step 2: Understand the Camera Perspective
- The camera is positioned at a lower angle, possibly from a vehicle, and is facing forward along the road.
- **The objects are positioned along the road, with the bus being farther away and the car being closer to the camera.**
Step 3: Interpret Relative Positioning Terms
- **According to the task, \"forward\" means farther from the camera, and \"backward\" means closer to the camera.**
- **The blue rigid bus is farther from the camera compared to the gray car.**
Step 4: Analyze Object Positions Relative to the Camera
- **The gray car is closer to the camera than the blue rigid bus.**
- **Therefore, the blue rigid bus is more forward relative to the camera.**
Step 5: Consider Possible Scenarios and Visualize
- Visualizing the scene confirms that the gray car is in front of the blue rigid bus from the camera's perspective.
Step 6: Evaluate Given Options and Select the Best Answer
- Given the options, \"No\" is the correct answer because the blue rigid bus is not behind the gray car; it is more forward from the camera's perspective.
Step 7: Verify Accuracy and Provide Final Answer
- The reasoning aligns with the task requirements and the provided information.
</think>
<answer>**No**</answer> ✅

Figure 10: Example of Front/Behind Determination task.

## B    Template for Generating VQA Tasks

### Template for Yaw Angle Determination Task

Task Description:
The primary goal of this task is to identify the direction that the specified object is facing in the given image. The camera in the image is facing { }, and you need to analyze the object's orientation based on this reference.

Question:
Which direction is { } facing in the image?
Options: - { } - { } - { } - { }

### Template for Pixel Location Estimation Task

Task Description:
The primary goal of this task is to accurately identify and provide the coordinates of a specified object within a given image. Your task is to analyze the image, locate the object, and return its position in the form of coordinates [x, y].

Question:
Where is { } located in the image?

### Template for Depth Range Determination Task

Task Description:
The primary goal of this task is to estimate the vertical distance of the specified object in the image from the camera, which is positioned at the origin. You need to analyze the image and choose the correct range of distance from the camera based on the visual cues provided.

Question:
How far is the vertical distance of { } in the picture from the camera?
Options: - { } - { } - { }

### Template for Distance Estimation Task

Task Description:
The primary goal of this task is to determine which of the two objects is closer to the camera that captured the image below. You need to assess the relative distance between the two objects based on the camera's perspective.

Question:
Which object, { } or { }, is { } to the camera?
Options: - { } - { } - Almost the same

### Template for Left/Right Determination Task

Task Description:
The primary goal of this task is to determine the relative left-right positioning of the two objects from the camera's perspective.

Question:
Which is further { }, { } or { }?
Options: - { } - { } - Almost the same

**Template for Front/Behind Determination Task**

Task Description:
The primary goal of this task is to determine the relative front-back positioning of the two objects from the camera's perspective, where the object farther from the camera is considered to be more forward.

Question:
Is {} {} {}?
Options: - Yes - No - Almost the same in terms of front-back position

## C Structured Response Format for the VQA Task

**Structured Response Format with Location Tag**

First, detect and list the target objects mentioned in the question from the image. Output their bounding boxes under the <location> tag. Next, reason carefully and step-by-step under the <think> tag to ensure logical accuracy and robustness, including any relevant error checks. Finally, provide a concise and definitive response in the <answer> tag.

Use the following format:
<location>[[Detected target object]: [xmin, ymin, xmax, ymax], ...]</location>
<think>[Step-by-step reasoning with attention to detail and potential error checks]</think>
<answer>[Final answer]</answer>

**Structured Response Format without Location Tag**

Reason carefully and step-by-step under the <think> tag to ensure logical accuracy and robustness, including any relevant error checks. Finally, provide a concise and definitive response in the <answer> tag.

Use the following format:
<think>[Step-by-step reasoning with attention to detail and potential error checks]</think>
<answer>[Final answer]</answer>

## D Prompts for High-Quality Chain-of-Thought Generation

**Prompt for Generating Chain-of-Thought**

Analyze the following task step by step to derive the best possible answer.
Task: {task}
Answer: {answer}
Please provide a detailed reasoning process, verify its accuracy, and then give your final answer clearly.

**Prompt for Summarizing Rules from Examples**

You are given the following reasoning examples. Analyze these examples to identify the underlying, generalizable problem-solving principles.
Examples: {examples}
Present your findings as bullet points in this format:
- Step 1: [core principle] - Step 2: [core principle] ...
Ensure these rules can be applied broadly to similar questions.

**Prompt for Generating Answers Using Extracted Rules**

Use the following principles to answer the question:
{rules}
Question: {question} Answer: {answer}
Provide a concise solution with key reasoning steps in the following format: <think>[Your step-by-step reasoning]</think> <answer>[Final answer]</answer>

**Prompt for Verifying and Refining Reasoning and Answers**

{response}
Evaluate the structured response above for logical consistency and completeness. Specifically:
1. Does the reasoning in <think> logically support the conclusion in <answer>?
2. Are there any internal contradictions, logical errors, or missing steps in the reasoning?
3. Is the reasoning chain complete and valid?
Provide your evaluation in the following format:
<reason>[A concise justification of your assessment or a brief note confirming the reasoning's validity]</reason> <validation>Valid / Invalid</validation>
Then, regardless of validity, output the full response in the following format:
- Keep <answer> unchanged.
- Modify <think> only if necessary to ensure logical soundness.
<think>[final version of reasoning steps]</think>
<answer>[original final answer]</answer>

