# OpenReview forum: "SURDS: Benchmarking Spatial Understanding and Reasoning in Driving Scenarios with Vision Language Models"
_NeurIPS.cc/2025/Datasets_and_Benchmarks_Track — NeurIPS 2025 Datasets and Benchmarks Track poster_

### Official Review · Reviewer_Q7NV · 2025-06-22

**Rating:** 5
**Confidence:** 4

**Summary:**

This paper introduces SURDS, the first benchmark specifically designed to systematically evaluate the spatial reasoning capabilities of VLMs. Built upon the nuScenes dataset, SURDS comprises 41,080 vision–question–answer training instances and 9,250 evaluation samples, spanning six spatial categories: orientation, depth estimation, pixel-level localization, pairwise distance, lateral ordering, and front–behind relations. The authors use SURDS to assess several state-of-the-art general-purpose VLMs, revealing significant limitations in their spatial understanding. To address these challenges, the paper proposes a reinforcement learning approach that incorporates positional and logical rewards, thereby enhancing the spatial reasoning abilities of VLMs. The proposed method achieves superior performance across comprehensive evaluation metrics, outperforming all existing mainstream VLMs.

**Dataset Code Accessibility:**

Yes

**Dataset Code Comments:**

The link provided in the paper points to a publicly accessible GitHub repository, whose README offers detailed instructions on downloading the datasets and reproducing the benchmark.

**Ethical Comments:**

The authors rely on publicly available data. For reproducibility, the authors provide open-source code and data.

**Ethical Considerations:**

No, there are no or only very minor ethics concerns

**Limitations Weaknesses:**

1. The manuscript would benefit from providing the complete model reasoning process in the appendix, thereby improving the overall persuasiveness and transparency of the work.
2. The writing quality requires further refinement; for example, redundant use of terms such as "vision–question–answer (VQA)" in Section 3 detracts from the clarity and conciseness of the text.
3. The formatting of the tables exhibits issues, including incomplete or broken lines, which detract from the professional presentation and overall quality of the paper.
4. The description of the training settings is not very clear in this version; for example, the base model used for training is not mentioned in the main text, even though it is shown in the table.

**Strengths Contributions:**

It is a good paper that highlights the limitations of current VLMs in spatial understanding by introducing the SURDS benchmark. It could contribute to the field by proposing a reward design aimed at improving the spatial reasoning abilities of these models. The benchmarks and datasets are well designed and could be used by follow-up researchers. The proposed reward function with both a localization and a logic reward could achieve good performance through GRPO training.
1. The data filtering pipeline is well-designed, and the benchmark is constructed thoughtfully to effectively evaluate the spatial reasoning capabilities of VLM.
2. The proposed method for generating CoT data is novel, offering the community a reliable CoT generation approach based on open-source models.
3. This work introduces a localization reward and a logic reward to enhance the spatial reasoning performance of VLMs. Both components are empirically validated through extensive experiments, demonstrating their efficacy.

---

> ### Author Rebuttal · Authors · 2025-07-30
>
> Dear Reviewer Q7NV:
>
> Thank you for your positive evaluation of our work and for raising insightful questions. We are pleased to hear that you found our benchmark design to be well-constructed. We also greatly appreciate your recognition of our approach to data chain-of-thought generation as well as the design of our location and logical rewards. Your encouraging feedback is highly valued and motivates us to further refine our contributions.
>
>
>
> **W1: The manuscript would benefit from providing the complete model reasoning process in the appendix, thereby improving the overall persuasiveness and transparency of the work.**
>
> A1: In our original submission, we included one complete reasoning example for each of the six tasks in the section 1 of the supplementary material. We will also include these examples in the appendix of the final version. As image display is not supported here, we provide the textual content of one example below, and we kindly refer you to the supplementary material for the full details.
>
> ```text
> Question: Task Description:
> The primary goal of this task is to identify the direction that the specified object is facing in the given image. The camera in the image is facing North, and you need to analyze the object's orientation based on this reference.
> Question: Which direction is the white car facing in the image?
> Options: \n- North\n- South\n- East\n- West
> First, detect and list the target objects mentioned in the question from the image. Output their bounding boxes under the <location> tag. Next, reason carefully and step-by-step under the <think> tag to ensure logical accuracy and robustness, including any relevant error checks. Finally, provide a concise and definitive response in the <answer> tag. Use the following format: <location>[[Detected target object]: [xmin, ymin, xmax, ymax], ...]</location>
> <think>[Step-by-step reasoning with attention to detail and potential error checks]</think><answer>[Final answer]</answer>
>
> Answer:
> <location>white car: [789, 519, 895, 601]</location>
> <think>
> Step 1: Establish a Reference Direction
> The camera is facing North, so this is our baseline direction.
> Step 2: Determine Object Position Relative to Camera
> The white car is positioned in front of the camera on the road, moving towards the camera.
> Step 3: Analyze Object Orientation Based on Reference and Position
> Since the car is moving towards the camera and the camera is facing North, the car must be facing South.
> Step 4: Consider Additional Contextual Clues
> There are no visible contextual clues such as road markings or traffic signs that would suggest a different orientation for the car.
> Step 5: Verify Consistency with Given Options
> The derived orientation of the car facing South is consistent with the given options.
> Step 6: Formulate a Clear Final Answer
> After verifying the analysis, the final answer is clear.
> </think>
> <answer>South</answer>
> ```
>
>
>
> **W2: The writing quality requires further refinement; for example, redundant use of terms such as "vision–question–answer (VQA)" in Section 3 detracts from the clarity and conciseness of the text.**
>
> A2: Thank you for your the feedback. We have thoroughly reviewed the manuscript to correct instances of repeated or inconsistent terminology definitions. Additionally, we have conducted another comprehensive quality check of the writing to ensure that similar issues will not appear in the final version.
>
>
>
> **W3: The formatting of the tables exhibits issues, including incomplete or broken lines, which detract from the professional presentation and overall quality of the paper.**
>
> A3: Thank you for your advice. We have corrected the issue of incomplete table lines and ensured that the tables in the final version are presented with improved visual quality.
>
>
>
> **W4: The description of the training settings is not very clear in this version; for example, the base model used for training is not mentioned in the main text, even though it is shown in the table.**
>
> A4: Thank you for your suggestion. We have revised the description of the training setup to ensure greater clarity and to avoid any potential ambiguity. The updated description of the training configuration is as follows:
>
> ```text
> Our data are sourced from the nuScenes dataset, which includes LidarSeg annotations. The threshold for the Labeled Object Overlap Filter Image is set to 0.8, while the threshold for the Unlabeled Object Overlap Filter Image is set to 0.4. We evaluate a variety of models, including state-of-the-art open-source and proprietary models, as well as models specifically designed for spatial understanding. A random baseline is also included for comparison, and the evaluated models are summarized in Table 2. All models are prompted with standardized instructions and are required to generate outputs that strictly adhere to a predefined format. We employ the sglang framework to accelerate inference and reduce evaluation time. All our training is conducted on Qwen2.5-VL-3B. All training and evaluation are performed on eight NVIDIA A800 GPUs. For supervised fine-tuning, models are trained for 2 epochs with a learning rate of 10^-6 and a warm-up ratio of 0.1. GRPO training is conducted for 1 epoch using a maximum prompt length of 4096 tokens, an output length of up to 512 tokens, and generating 4 samples per prompt. The structured response format is provided in Appendix C.
> ```

---

> > ### Comment · Reviewer_Q7NV · 2025-08-07
> >
> > Thank you for the author's reply. l will keep the Accept rating unchanged.

---

> > > ### Author Response · Authors · 2025-08-08
> > >
> > > Thanks very much for the recgonition of our work from the reviewer. We will refine based all the thoughtful comments for our final version.

---

### Official Review · Reviewer_vjce · 2025-06-25

**Rating:** 4
**Confidence:** 3

**Summary:**

This paper proposes a large-scale benchmark designed to systematically evaluate the spatial reasoning capabilities of VLMs in driving scenarios that span six spatial reasoning categories, which exposes persistent shortcomings in fine-grained spatial understanding. Based on this, a reinforcement-learning framework is proposed to mitigate the deficiencies. Extensive experiments are conducted to verify the performance on existing VLMs. Besides, the efficacy of the training method and the reward design can be verified on the experiments and ablations

**Dataset Code Accessibility:**

Yes

**Ethical Considerations:**

No, there are no or only very minor ethics concerns

**Final Justification:**

My concerns have been addressed mostly. The final rating is set to 4.

**Limitations Weaknesses:**

1.	SFT and RL with reward model are proven effective methods for VLM or LLM post-training. More specific designs and rewards can be proposed for the characteristics of driving scenarios.
2.	For the driving scenarios, accurate special reasoning and understanding can be essential for further trajectory prediction and planning. Thus, more practical QA pairs or those related to downstream tasks can be more valuable for benchmark estimation.
3.	The evaluation metrics are too simplistic. Solely 0 or 1 is not suitable for the determination of yaw angle, depth, and distance, which require more precise evaluation to verify the models’ performance.
4.	As a paper for the benchmark track, a comparison of the scale of datasets and the kinds of annotations or QA pairs of the proposed benchmark with previous benchmarks is required to verify the significance and novelty of this new study.

**Strengths Contributions:**

1.	This paper is well-written and easy-to-follow. Figures are clear to express the design and ideas.
2.	This benchmark specifically targets crucial spatial understanding in driving scenarios. The six well-motivated spatial tasks provide a holistic assessment of spatial understanding.
3.	The reinforcement learning strategy is specifically designed to mitigate the deficiencies. Extensive experiments are conducted to expose the problems and verify the effectiveness of the design.

---

> ### Author Rebuttal · Authors · 2025-07-30
>
> Dear Reviewer vjce:
> We sincerely thank you for the time and effort you dedicated to reviewing our paper. We are pleased to hear your positive recognition of the effectiveness of our proposed reinforcement learning strategy. Your comprehensive feedback on our reward design, benchmark QA construction and evaluation metrics, as well as the dataset scale and annotation types, is highly encouraging. We hope that our responses below will address your concerns and help clarify the contributions of our work, and we kindly ask you to consider updating the score accordingly.
>
>
>
> **W1: SFT and RL with reward model are proven effective methods for VLM or LLM post-training. More specific designs and rewards can be proposed for the characteristics of driving scenarios.**
>
> A1: To the best of our knowledge, there has been no prior work specifically focusing on reward design for spatial reasoning tasks. We are **the first to propose location and logical rewards tailored for spatial reasoning**, designed to be general with the goal of drawing the community’s attention and providing valuable insights. Our current design has already shown positive experimental results and has received positive recognition from other reviewers. We believe that the absence of specialized reward function exploration does not reduce the merit of our work.
>
>
>
> **W2: For the driving scenarios, accurate special reasoning and understanding can be essential for further trajectory prediction and planning. Thus, more practical QA pairs or those related to downstream tasks can be more valuable for benchmark estimation.**
>
> A2: Thanks for the feedback. Please note that our current benchmark is **designed to evaluate and boost the spatial/perception reasoning in VLMs under a driving scenario**, similar to prior studies [1–4]. **Trajectory prediction and planning** are normally treated as abilities after the spatial perception, which is **beyond the scope of this work** and will be our future work.
>
> However, our work does have potential for easy extension. The annotations we provide consist of independent information for each object and its corresponding image, enabling researchers to easily construct downstream QA pairs based on our annotations.
>
> [1] Fu, Xingyu, et al. "Blink: Multimodal large language models can see but not perceive." European Conference on Computer Vision. Cham: Springer Nature Switzerland, 2024.
>
> [2] Cheng, An-Chieh, et al. "Spatialrgpt: Grounded spatial reasoning in vision-language models." Advances in Neural Information Processing Systems 37 (2024): 135062-135093.
>
> [3] Cai, Wenxiao, et al. “SpatialBot: Precise Spatial Understanding with Vision Language Models.” Proceedings of the 2025 IEEE International Conference on Robotics and Automation. 2025.
>
> [4] Yang, Jihan, et al. "Thinking in space: How multimodal large language models see, remember, and recall spaces." Proceedings of the Computer Vision and Pattern Recognition Conference. 2025.
>
>
>
> **W3: The evaluation metrics are too simplistic. Solely 0 or 1 is not suitable for the determination of yaw angle, depth, and distance, which require more precise evaluation to verify the models’ performance.**
>
> A3: **Our design logic is to evaluate VLMs from a human spatial reasoning perspective, more specifically, relative spatial relations.** We convert these relative spatial relations into categorical ranges rather than precise scalars to aquire a honest benchamrk . Our benchmark asks models to infer spatial attributes by combining visual cues with prior knowledge, thereby assessing whether they achieve human-like understanding of spatial relations. For instance, a person can usually decide whether an object in a single image faces roughly north, south, east, or west, but cannot quote its exact yaw angle—illustrating that spatial judgments in the image are expressed as ranges, not precise numbers.
>
> **Directly evaluating exact numerical values from 2-D imagery yields misleading metrics and has trival solution for model to hack.** In early trials—including a vertical-distance task—we evaluated absolute quantities, but the setting proved impractical: precise numbers (e.g., yaw angles) are inherently hard to recover from a single view, and, even with Mean Relative Accuracy (MRA) [1], the resulting evaluation bias can only be partly alleviated. Worse, a model could **hallucinate by repeating a single average value for every query** and still post an apparently decent score, masking its true spatial-reasoning ability. We provide several key results from our previous work below for your reference.
>
> |      Model       | Relative Horizontal Distance | Relative Distance | Camera Vertical Distance | Camera Distance |
> | :--------------: | :--------------------------: | :---------------: | :----------------------: | :-------------: |
> |      GPT-4o      |             9.58             |       14.69       |          46.11           |      70.65      |
> | Gemini-1.5-flash |            24.16             |       69.60       |          65.17           |      67.92      |
> |  LLavVA-ov-72b   |            13.76             |       45.27       |          42.97           |      27.09      |
>
> **We therefore employ a dual-query, range-based evaluation that curbs hallucination-induced score inflation and better reflects genuine reasoning.** For each item, we pose two complementary questions with opposite perspectives, and count the item correct only when both answers are right. This design sharply reduces metric inflation, increases robustness, and delivers a more reliable assessment of a VLM’s spatial-reasoning capacity.
>
> [1] Yang, Jihan, et al. "Thinking in space: How multimodal large language models see, remember, and recall spaces." Proceedings of the Computer Vision and Pattern Recognition Conference. 2025.
>
>
>
> **W4: As a paper for the benchmark track, a comparison of the scale of datasets and the kinds of annotations or QA pairs of the proposed benchmark with previous benchmarks is required to verify the significance and novelty of this new study.**
>
> A4: In Table 1 of our paper, we already provide a detailed comparison with prior benchmarks. Because of space constraints, explicit columns for benchmark scale and annotation type were omitted; we now supply these statistics below and will include them in the final version. As the expanded table shows, our benchmark retains a clear scale advantage over existing work.
>
> |             Paper             | Benchmark Scale |  Annotations Types  | Data Source | Reasoning | Method | w/o Depth |    w/o Visual Mark    |
> | :---------------------------: | :-------------: | :-----------------: | :---------: | :-------: | :----: | :-------: | :-------------------: |
> |     BLINK (ECCV2024) [1]      |      3807       | Image-Text QA Pairs |     Web     |     ×     |   ×    |     √     |    w/ Marked point    |
> | SpatialRGBT (NeurIPS2024) [2] |      1406       | Image-Text QA Pairs |     Web     |     √     |   √    |     ×     |        w/ Mask        |
> |   SpatialBot (ICRA2025) [3]   |       174       | Image-Text QA Pairs |     Web     |     √     |   √    |     ×     | w/ Marked point &Bbox |
> |   VSI bench (CVPR2025) [4]    |     over 5k     | Video-Text QA Pairs |   Indoor    |     √     |   ×    |     √     |           √           |
> |          SURDS(ours)          |      9250       | Image-Text QA Pairs |   Driving   |     √     |   √    |     √     |           √           |
>
> [1] Fu, Xingyu, et al. "Blink: Multimodal large language models can see but not perceive." European Conference on Computer Vision. Cham: Springer Nature Switzerland, 2024.
>
> [2] Cheng, An-Chieh, et al. "Spatialrgpt: Grounded spatial reasoning in vision-language models." Advances in Neural Information Processing Systems 37 (2024): 135062-135093.
>
> [3] Cai, Wenxiao, et al. “SpatialBot: Precise Spatial Understanding with Vision Language Models.” Proceedings of the 2025 IEEE International Conference on Robotics and Automation. 2025.
>
> [4] Yang, Jihan, et al. "Thinking in space: How multimodal large language models see, remember, and recall spaces." Proceedings of the Computer Vision and Pattern Recognition Conference. 2025.

---

> > ### Comment · Reviewer_vjce · 2025-08-04
> >
> > My concerns have been addressed mostly, especially regarding the explanation in the evaluation metrics. I will raise my score from 3 to 4. Additionally, what needs to be considered is: if the determination of yaw angle, depth, and distance is roughly in a range and not a precise value, it requires a more precise definition for the task and its classification. Thanks for your feedback.

---

> > > ### Author Response · Authors · 2025-08-08
> > >
> > > We sincerely thank the kind reviewer for their thoughtful comments and valuable suggestions, which have greatly contributed to enhancing the quality of our paper.

---

### Official Review · Reviewer_o2yu · 2025-06-27

**Rating:** 5
**Confidence:** 4

**Summary:**

The paper proposes SURDS, the first large-scale benchmark for evaluating the fine-grained spatial reasoning capabilities of vision-language models (VLMs) specifically for autonomous driving scenarios. The benchmark is based on nuScenes multi-view data, and 41,080 training and 9,250 validation VQA samples are constructed after multi-stage occlusion, size, and ambiguity filtering, covering six types of tasks: orientation, depth, pixel-level localization, distance, left-right order, and front-back relationship. The authors systematically test cutting-edge VLMs such as GPT-4o, Gemini 2.0 Flash, and Qwen-2.5-VL, and find that they still have significant shortcomings in fine-grained spatial understanding. To improve the capability, the paper proposes a GRPO-enhanced learning alignment method combining position reward + logical consistency reward, which enables the 3B-parameter Qwen-2.5-VL model to achieve a score of 40.80 on SURDS, significantly outperforming the performance of large models such as GPT-4o (13.30) and Gemini 2.0 Flash (35.71).

**Dataset Code Accessibility:**

Yes

**Dataset Code Comments:**

The code and data can be accessed normally.

**Ethical Considerations:**

No, there are no or only very minor ethics concerns

**Final Justification:**

The authors couldn’t test larger models due to time and compute limits, which is understandable. They cited relevant studies to explain the stability of their reward design and how their data filtering could transfer, but the paper still lacks direct tests on larger models and on pure RGB data. Considering all this, I maintain my current score.

**Limitations Weaknesses:**

1. Reinforcement learning alignment is validated only on a 3 B–parameter model; the authors also admit they have not tested its applicability on larger parameter scales or the effectiveness of linear reward scaling strategies.
2. The paper notes that using a reward range of (–1, 1) causes gradients to remain negative or near zero for extended periods, indicating that the current design is still sensitive to reward density and stability, which may limit extension to more complex scenarios.
3. Data filtering relies on LiDAR point clouds to verify visibility; however, the evaluation tasks themselves only provide RGB data. If SURDS were migrated to a dataset without depth sensors, the filtering strategy and labeling consistency might be insufficient.

There are no other obvious issues.

**Strengths Contributions:**

1. SURDS is the first benchmark for spatial reasoning in autonomous driving scenarios, featuring a large dataset and multidimensional tasks that surpass previous works focused solely on indoor or 2D relationships.
2. It employs label + text dual filtering and LiDAR projection verification to ensure targets are clearly visible and descriptions unambiguous, thereby improving evaluation reliability.
3. It incorporates spatial IoU and reasoning–answer consistency into the reward design, alongside format and answer correctness, to drive GRPO training; experiments show that the synergy of these four rewards significantly enhances spatial reasoning performance.
4. The dataset, evaluation tools, and alignment code are publicly released.

---

> ### Author Rebuttal · Authors · 2025-07-30
>
> Dear Reviewer o2yu:
> We are pleased that you found our data filtering approach effective in enhancing evaluation reliability and appreciated the proposed benchmark. We also thank you for recognizing the novelty and effectiveness of our training methodology, particularly the use of location and logic rewards to strengthen the model’s spatial reasoning capabilities. We sincerely appreciate your positive feedback on our work.
>
>
>
> **W1: Reinforcement learning alignment is validated only on a 3 B–parameter model; the authors also admit they have not tested its applicability on larger parameter scales or the effectiveness of linear reward scaling strategies.**
>
> A1: Thank you for your insightful suggestion. Our experiments were conducted on a 3 B-parameter model and already show positive results. Crucially, the training methodologies we employed were not hand-crafted for this specific model size. Prior work—such as Visual-RFT [1]—shows that, after training, mAP improved by 15.4 for Qwen2-VL-2B and by 18.4 for Qwen2-VL-7B, indicating that **the same methodology transfers well to larger models** (e.g., 7 B parameters and beyond).
>
> However, due to limitations in time and computational resources, we were unable to conduct experiments on larger models within the rebuttal period. We plan to further explore and validate this conclusion on larger-scale models as part of our future work.
>
> [1] Liu, Ziyu, et al. “Visual-RFT: Visual Reinforcement Fine-Tuning.” Proceedings of the IEEE/CVF International Conference on Computer Vision (ICCV), 2025.
>
>
>
> **W2: The paper notes that using a reward range of (–1, 1) causes gradients to remain negative or near zero for extended periods, indicating that the current design is still sensitive to reward density and stability, which may limit extension to more complex scenarios.**
>
> A2: Please note, this does not limit its extension to more complex scenarios. The additional experiments with the (–1, 1) range were included to highlight the importance of reward range selection, particularly for complex tasks where reward is sparse. In our main experiments, we also adopted the (0, 1) reward range because the VQA tasks in our proposed benchmark are highly challenging for current VLMs, leading to sparse rewards during training. Using the (0, 1) range helps prevent the positive rewards from being diluted.
>
> We would like to clarify that the GRPO algorithm is inherently sensitive to reward density, which is why the conventional reward range is typically set to (0, 1) without negative rewards, as used in VLM-R1 [1]. However, this does not imply that GRPO cannot be applied to more complex scenarios. For instance, Embodied-R [2] applies the same algorithm to an embodied drone-navigation task: after receiving a multi-step instruction such as “Observe around, fly toward the road, then turn left and land on the roof”, the agent must pick the statement that matches its current action (e.g., “I fly toward the road”) from several alternatives. The success of GRPO in this complex, sparse-reward environment demonstrates that **our methodology is not limited in its extension to more complex scenarios**.
>
> [1] Shen, Haozhan, et al. "Vlm-r1: A stable and generalizable r1-style large vision-language model." arXiv preprint arXiv:2504.07615 (2025).
>
> [2] Zhao, Baining, et al. "Embodied-R: Collaborative Framework for Activating Embodied Spatial Reasoning in Foundation Models via Reinforcement Learning." arXiv preprint arXiv:2504.12680 (2025).
>
>
>
> **W3: Data filtering relies on LiDAR point clouds to verify visibility; however, the evaluation tasks themselves only provide RGB data. If SURDS were migrated to a dataset without depth sensors, the filtering strategy and labeling consistency might be insufficient.**
>
> A3: Although our benchmark follows the standard spatial-reasoning task settings for VLMs [1-3]—hence the evaluation itself consumes only RGB inputs—the underlying data-filtering methodology is readily transferable.
>
> We chose to use the LiDAR point clouds and depth annotations from nuScenes [4] for data filtering because its annotations are highly comprehensive. However, LiDAR data in our pipeline is used solely to determine whether an object is occluded and thus not visible—a function that can also be effectively achieved through image-based features when LiDAR is unavailable.
>
> For densely annotated datasets without LiDAR (e.g., Habitat [5]), our data-filtering strategy can be adapted by leveraging models such as object detection and instance segmentation to perform occlusion filtering.
>
> We emphasize that data quality is more critical than quantity. Constructing spatial reasoning datasets without reliable depth information is labor-intensive and offers limited benefits, as depth generated by alternative means cannot be robustly validated.
>
> [1] Fu, Xingyu, et al. "Blink: Multimodal large language models can see but not perceive." European Conference on Computer Vision. Cham: Springer Nature Switzerland, 2024.
>
> [2] Cheng, An-Chieh, et al. "Spatialrgpt: Grounded spatial reasoning in vision-language models." Advances in Neural Information Processing Systems 37 (2024): 135062-135093.
>
> [3] Cai, Wenxiao, et al. “SpatialBot: Precise Spatial Understanding with Vision Language Models.” Proceedings of the 2025 IEEE International Conference on Robotics and Automation. 2025.
>
> [4] Caesar, Holger, et al. "nuscenes: A multimodal dataset for autonomous driving." Proceedings of the IEEE/CVF conference on computer vision and pattern recognition. 2020.
>
> [5] Puig, Xavier, et al. "Habitat 3.0: A co-habitat for humans, avatars and robots." arXiv preprint arXiv:2310.13724 (2023).

---

> > ### Comment · Reviewer_o2yu · 2025-08-01
> >
> > Thank you for the author's reply. l will keep the Accept rating unchanged.

---

> > > ### Author Response · Authors · 2025-08-08
> > >
> > > We would like to sincerely thank the review for their comments. We will refine our final version based on your thoughtful feedback.

---

### Note · Authors · 2025-08-14

Dear Area Chair and Reviewers,

We sincerely thank you for your constructive comments, thoughtful engagement, and the time devoted to evaluating our manuscript. Your feedback substantially improved the clarity and quality of the paper. We are encouraged that our rebuttal addressed key concerns, and that some reviewers subsequently raised their scores. The main outcomes from the discussion phase are:

`Reviewer o2yu` maintained an “**Accept**” recommendation, acknowledging our training approach’s scalability across model sizes and scenario complexity, as well as the adaptability of our data-filtering strategy.

`Reviewer vjce` **raised the score** to “**Borderline Accept**,” acknowledging our clarifications, the novelty of our location and logical rewards, the advantages of our proposed benchmark over prior work, and a deeper understanding of our task setup.

`Reviewer Q7NV` maintained an “**Accept**” recommendation and recognized the value of the complete examples provided in the supplementary materials. We have incorporated their suggestions to improve readability and presentation.

We conclude by reiterating our contributions: We propose SURDS, **the first large-scale benchmark for evaluating fine-grained spatial understanding of VLMs in realistic driving scenarios**. Our evaluations on SURDS reveal fundamental spatial reasoning limitations in existing models and demonstrate that model scale alone does not ensure spatial competence. Comprehensive experiments investigated different training strategies from supervised fine-tuning to reinforcement learning post-train alignment, providing valuable reference for follow-up researchers.

Once again, we deeply appreciate the Area Chair’s and reviewers’ time and effort, and we believe the discussion has significantly strengthened the paper.

With sincere respect and gratitude,

The Authors

---

### Decision · Program_Chairs · 2025-09-18

**Decision:**

Accept (poster)

**Comment:**

The paper introduces SURDS, the first large-scale benchmark specifically designed to systematically evaluate the fine-grained spatial reasoning capabilities of vision–language models in autonomous driving scenarios, spanning six reasoning categories and revealing persistent shortcomings in spatial understanding. The initial evaluations from the three reviewers were divided, with two accepts and one borderline reject. The reviewer who gave the borderline reject noted the need for more specific designs and rewards, more practical QA pairs, a more sophisticated evaluation metric, and a more detailed comparison with existing benchmarks. However, they stated that these concerns were mostly addressed in the authors’ rebuttal and raised their final score to borderline accept. Consequently, all reviewers gave positive evaluations in the end (one borderline accept and two accepts), and the AC recommends acceptance of this paper. That said, it is worth noting that the reviewers’ feedback and final justifications still include comments for further improvement, and it is desirable that appropriate revisions be incorporated into the final version if possible.